# Novel perovskite solar cell with Distributed Bragg Reflector

Waqas Farooq[1], Shanshan Tu[2]*, Syed Asfandyar Ali Kazmi[1], Sadaqat ur Rehman[3], Adnan Daud Khan[4], Haseeb Ahmad Khan[5], Muhammad Waqas[2], Obaid ur Rehman[1], Haider Ali[6], Muhammad Noman[4]

1 Department of Electrical Engineering, Sarhad University of Science & IT, Peshawar, Pakistan,
2 Engineering Research Center of Intelligent Perception and Autonomous Control, Faculty of Information Technology, Beijing University of Technology, Beijing, China, 3 Department of Computer Science, Namal Institute, Mianwali, Pakistan, 4 US-Pakistan Center for Advanced Studies in Energy, University of Engineering & Technology, Peshawar, Khyber Pukhtunkhwa, Pakistan, 5 Department of Electrical Engineering, University of Engineering and Technology, Mardan, Khyber Pukhtunkhwa, Pakistan, 6 Department of Electrical and Electronics Engineering Technology, University of Technology, Nowshera, Pakistan

* sstu@bjut.edu.cn

**Data Availability Statement:** Data is available in the supporting information as S1 Dataset.

**Funding:** This work was sponsored through grants awarded to ST by Chongqing Industrial Control System Security Situational Awareness Platform, 2019 Industrial Internet Innovation and

## Abstract

This paper reports numerical modeling of perovskite solar cell which has been knotted with Distributed Bragg Reflector pairs to extract high energy efficiency. The geometry of the proposed cells is simulated with three different kinds of perovskite materials including $CH_3NH_3PbI_3$, $CH_3NH_3PbBr_3$, and $CH_3NH_3SnI_3$. The toxic perovskite material based on Lead iodide and lead bromide appears to be more efficient as compared to non-toxic perovskite material. The executed simulated photovoltaic parameters with the highest efficient structure are open circuit voltage = 1.409 (V), short circuit current density = 24.09 mA/cm², fill factor = 86.18%, and efficiency = 24.38%. Moreover, a comparison of the current study with different kinds of structures has been made and surprisingly our novel geometry holds enhanced performance parameters that are featured with back reflector pairs (Si/SiO₂). The applied numerical approach and presented designing effort of geometry are beneficial to obtain results that have the potential to address problems with less efficient thin-film solar cells.

## Introduction

Substitute to fossil fuel to generate energy through cleaner source is solar energy which is an endless and unlimited source for obtaining sunlight that can be transformed into another form of energy. Different approaches [1, 2], techniques [3–5], and methods [6, 7] have been investigated, demonstrated, and reported to improve the design and geometry of the device that can be used to generate electricity and can be utilized as a replacement to typical non-environmental friendly methods [8] that have been used to generate power. Solar cell technology is an inexhaustible, reliable, and commercialized technology that has been considered by the photovoltaic community to generate electric power through the photovoltaic effect [9, 10].

Development Project – Provincial Industrial Control System Security Situational Awareness Platform, Beijing Natural Science Foundation (No. 4212015), Natural Science Foundation of China (No. 61801008), China Ministry of Education - China Mobile Scientific Research Foundation (No. MCM20200102), China Postdoctoral Science Foundation (No. 2020M670074), and the Beijing Municipal Commission of Education Foundation (No. KM201910005025).

**Competing interests:** The authors have declared that no competing interests exist.

Different materials such as organic polymers [11, 12], silicon [13, 14], CIGS [15, 16], and CdS/CdTe [17, 18] have been investigated numerically to improve the device performance to obtain high conversion efficiency. Moreover, numerical modeling and simulation are always encouraged to estimate the parameters before moving towards the fabrication side. The benefit of numerically modeling before fabrication helps to avoid unwanted results and save time as well as manufacturing cost [19, 20]. Several different kinds and materials have been synthesized [21, 22] and investigated to produce high conversion efficiency. However, the hindrance to commercializing these technologies on large scale is still a challenging task because of low efficiency which restricts the usage of this thin-film technology. Major losses in the solar cells mostly occur due to reflection and utilization of that architecture which does not have sufficient capacity to absorb sunlight. On the other hand, cell architecture which supports a large number of losses needs kind can be of great change if an efficient design is made such as the utilization of different methods and techniques. On top of that, Distributed Bragg Reflector (DBR) [23, 24], found to be an imperative approach to absorb the light by reflecting those lights which were supposed to be a loss but can be capture by using different pairs with relevant materials. Management to capture the reflected light from the back surface towards the active region is one of the most important and difficult parameters as it requires a high beam of approach to reduce the optical losses by utilizing different imperative materials which significantly results in the improvement of the device performance and provide high efficiency.

J. Duan et al., investigated lanthanide ions doped CsPbBr3 Halides for HTM and obtain 10.14% efficiency [25]. T. Singh et al., fabricated perovskite solar cells in ambient air under controlled humidity and extracted 20.8% efficiency [26]. S. A. Kazmi et al., investigated cadmium telluride solar cell with three pairs of DBR and obtained an $\eta$ of 23.94% [24]. J. Feng et al., investigated stable flexible perovskite solar cells using an effective additive assistant strategy and extracted 22.7% efficiency [27]. J. J. Yoo et al., investigated an interface stabilized perovskite solar cell with low voltage loss and obtained an efficiency of 22.6% [28]. J. C. Yu et al., investigated stable inverted perovskite solar cells via treatment by semiconducting chemical additive and obtained 20.3% efficiency [29]. E. H. Jung et al., investigated stable and scalable perovskite solar cells using poly(3-hexylthiophene) and extracted 22.72% efficiency [30]. H. Ren et al., invested the stable Ruddlesden–Popper perovskite solar cell with tailored interlayer molecular interaction and obtained an efficiency of 18.06% [31]. X. Ren et al., investigated chlorine-modified SnO2 electron transport layer for high-efficiency perovskite solar cells and obtained an efficiency of 17.81% [32]. A. Solanki et al., reported efficiency of 20.83% by using a novel approach of heavy water additive in formamidinium for efficiency enhancement in perovskite solar cells [33]. J-H. Lee et al., utilized ($SnO_2$-$SiO_2$) as a DBR material in MAPbI$_3$ based perovskite solar cell and reported the highest conversion efficiency of 9.52% [34]. O. Isabella et al., improved the performance of the thin-film silicon solar cell by utilizing a-Si:H/SiNx:H as a DBR layer material and covered the reflectance peak of 600 nm [35]. Y. Peng et al., used $TiO_2$/$SiO_2$ as a DBR material and observed an enhancement of over 20% efficiency in the solar cell when fabricated on glass and PET substrate [36]. S. Mitra et al., numerically investigated the different combination of DBR materials such as $SiO_2$/a:Si, $SiO_2$/$TiO_2$ and $SiO_2$/SiNx via 3-FDTD simulation and optimized the structure for covering wavelength range of 900–1100 nm [37].

Herein, we numerically commutated thin-film perovskite solar cells with DBR pairs in different schematics. The structure based on methylammonium lead bromide was found to be more efficient as compared to other perovskite materials.

## Framework and modeling

Using the general purpose photovoltaic device model, the geometry of the solar cells is designed with different functional layers as depicted in Fig 1. The function of the functional layers is as follows: Glass as a protecting layer. FTO as a top transparent electrode, $Al_2O_3$ as hole blocking layer (HBL), the advantage of using HBL layer is that it strongly block the flow of holes in the upper region to avoid recombination, $SnO_2$ as an electron transport layer (ETL), as ETL provides a smooth path for the flow of electrons [38]. Next, the perovskite layer is patched as a major light-harvesting layer because perovskite can deliver high energy efficiency. Moreover, it has low manufacturing cost and holds high mechanical flexibility which makes it ideal for thin-film technology (TFT). Next to the active layer, the hole transport layer (HTL) is stacked to provide a path for the flow of holes [39]. For the HTL, Spiro OmeTAD is selected due to its tremendous properties such as high thermal stability, high mobility of holes, and easy manufacturing process. For the bottom electrode, zinc oxide (ZnO) is doped with aluminum (Al). The doped electrode helps in passing the light towards the DBR section which is composed of effective reflecting materials silicon (Si) and silicon dioxide ($SiO_2$), which is also known as stannic oxide. The DBR section consist of four pairs of Si/$SiO_2$ as a back reflector.

Numerically, the structure is based on the drift-diffusion model which can be represented by Eq 1 and Eq 2

$$J_n = q\mu_c n \frac{\partial E_c}{\partial x} + qD_n \frac{\partial n}{\partial x} \tag{1}$$

$$J_p = q\mu_c p \frac{\partial E_v}{\partial x} - qD_p \frac{\partial p}{\partial x} \tag{2}$$

The utilization of Poisson equation and equation of continuity for calculations can be expressed as Eq 3, Eq 4 and Eq 5 respectively.

$$\frac{d}{d_x} \varepsilon_o \varepsilon_r \cdot \frac{d_\varphi}{d_x} = q(n - p) \tag{3}$$

$$\frac{\partial J_n}{\partial x} = q\left(R_n - G + \frac{\partial n}{\partial t}\right) \tag{4}$$

$$\frac{\partial J_p}{\partial x} = q\left(R_p + G + \frac{\partial p}{\partial t}\right) \tag{5}$$

The fill factor and PCE of the device can be calculated by Eq 6 and Eq 7 respectively.

$$FF = \frac{J_{mp} V_{mp}}{J_{sc} V_{oc}} \tag{6}$$

$$PCE, \eta(\%) = \frac{v_{oc} \cdot lsc \cdot FF}{P_{input}} \times 100 \tag{7}$$

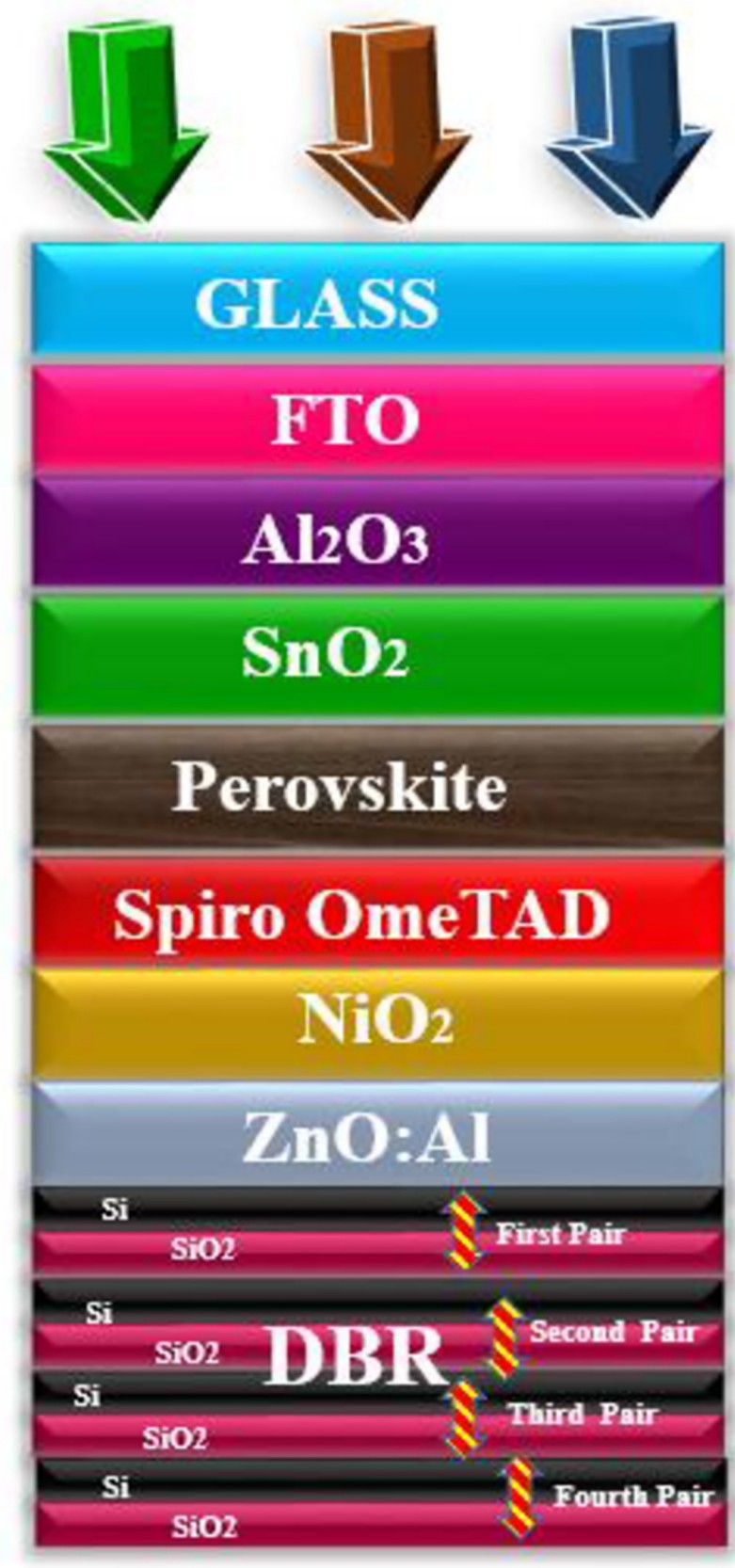

**Fig 1. Geometry of the proposed solar cell composed of different functional materials, glass as protecting layer, FTO as a top transparent electrode, Al₂O₃ as HBL, SnO₂ as ETL, Perovskite as an active layer, Spiro OmeTAD as HTL, NiO₂ as EBL, ZnO:Al as transparent electrode/light spectrum passing layer towards DBR pairs, and DBR as a back reflector (Si/SiO₂).**

## Results and discussion

Photons Management in a solar cell depends on the geometry of the cell in which it has been designed. The geometry of the cell must consist of those materials which can absorb sunlight in a bulk amount that can result in the enhancement of e-h pairs which further boost the performance of the cell. In this objective full study, the aim of achieving high conversion efficiency from perovskite solar cell is obtained by using a different number of pairs in DBR stacked. Three different cases with different perovskite materials are investigated with deep insight into the cell thickness to obtained high-performance parameters. The simulated input parameters are displayed in Table 1.

### Case 1

To extract high-performance parameters there is a dire need to utilize those materials which have a high coefficient of absorption. As the high absorption coefficient materials help to absorb the incoming photonic energy light more efficiently as compared to those which have a low absorption coefficient. Fig 2 shows the proposed geometry of the cell in which lead bromide perovskite is used as a major absorption layer and modulated between 500–600 nm to attain high electrical parameters. The cell performance increases with the increase in the thickness of the active layer. The observed improvement in the cell is because of the active layer which shows high absorption. The improved parameters at 570 nm is a shred of evidence that indicates that at this optimal thickness the absorption is high which sequentially delivers high PV parameters. This high absorption further helps in the creation of e-h pairs which mobilize in the material and got collected at the respected electrode after giving high values of the performance parameters. As the amount of the active layer thickness increases in the cell, the supporting parameters such as $V_{oc}$, $J_{sc}$, $FF$, and $PCE$ also increases as shown Fig 3(A)–3(D) respectively. The $V_{oc}$ of the device increases from 1.144–1.325 V when the thickness of the cell heightened from 500–570 nm whereas the $J_{sc}$ increases from 20.832–23.14 mA/cm$^2$, $FF$ from

**Table 1. Simulated input parameters.**

| Parameters | FTO | Al₂O₃ | SnO₂ | NiO₂ | Pb based Perovskite | ZnO:Al | Tin based Perovskite | Si | SiO₂ | Spiro |
|---|---|---|---|---|---|---|---|---|---|---|
| $m^*_n/m_o$ | 0.26 | 2.86 | 0.24 | 1.794 | - | 0.010 | - | 162 | - | - |
| $m^*_p/m_o$ | 0.6 | 4.23 | 0.4 | 1.78 | - | 3.37 | - | 1.124 | - | - |
| Dielectric Constant $\varepsilon/\varepsilon_0$ | 2.846 | 9.8 | 9.86 | - | - | 4.45 | - | 4.05 | 3.9 | 4.4 |
| Electron Affinity | 3.2 | 3.71 | 7.47 | 1.46 | - | 9 | 4.17 | 11.8 | 1.5 | 2.2 |
| Electron Mobility $\mu_e$ | 20E-4 | 165 | 23–106 | 32.54 | $2.33 \times 10^{-4}$ | 2.320E+18 | $1.6 \times 10^{-4}$ | 2.800E+19 | 20 | $1 \times 10^{-8}$ |
| Hole Mobility $\mu_h$ | 10E-4 | 5 | 6 | 0.07–4.4 | $3.22 \times 10^{10-4}$ | 1.845E+19 | $1.6 \times 10^{-4}$ | 2.600E+19 | 18 | $1 \times 10^{-8}$ |
| Band Gap Energy $E_g$ | 3.25 | 4.64 | 3.57 | 3.6 | 1.6 | 2.5 | 1.3 | 1.12 | 8.76 | 2.9 |
| Conduction band effective density of states NC | $1 \times 10^{22}$ | $1.50 \times 10^{18}$ | $1.04 \times 10^{19}$ | $3.2 \cdot 10^{19}$ | - | $1.2 \times 10^{17}$ | - | - | - | - |
| Valence band effective density of states NV | - | 1.80E+19 | $1.8 \times 10^{19}$ | - | - | - | - | 1.000E+19 | | - |

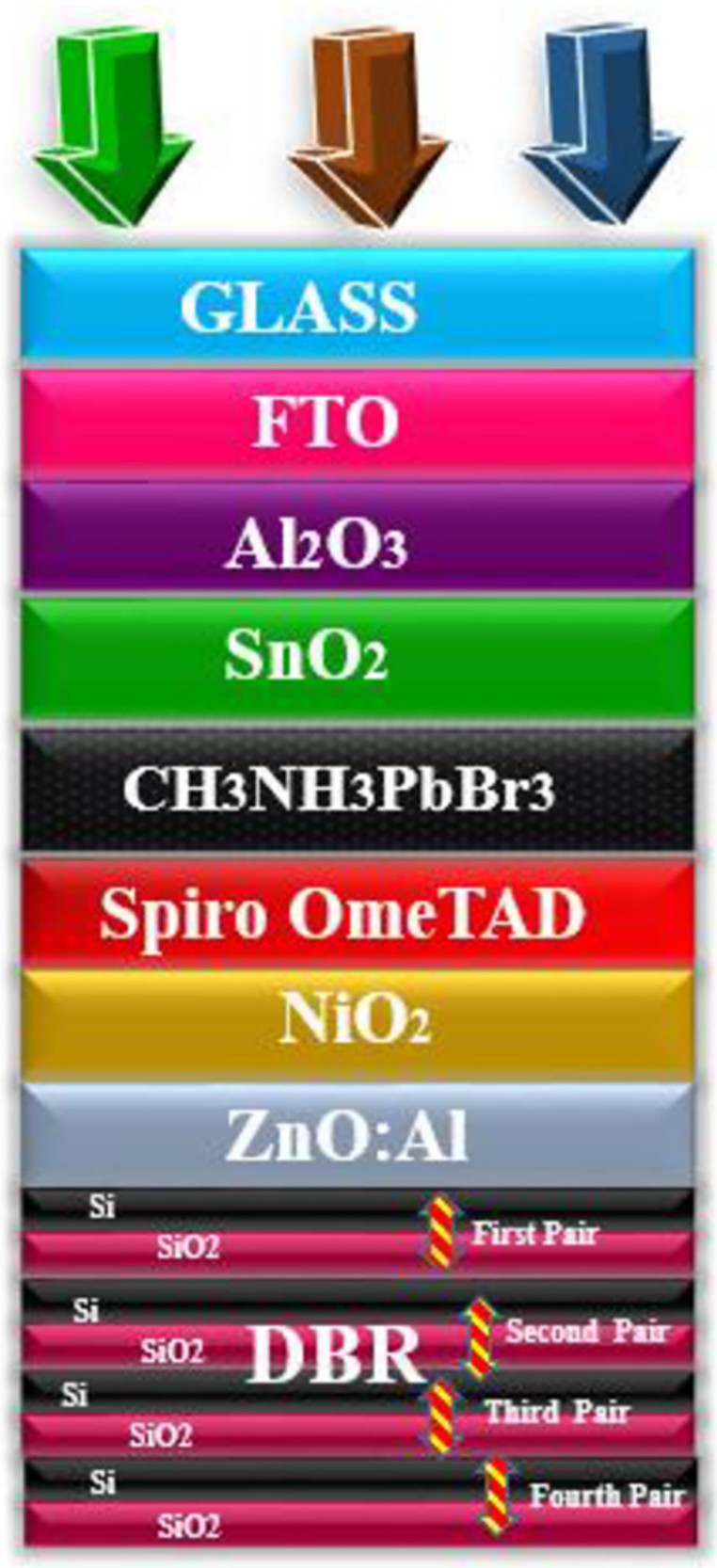

**Fig 2. Geometry of the proposed solar cell composed of different functional materials, glass as protecting layer, FTO as a top transparent electrode, $Al_2O_3$ as HBL, $SnO_2$ as ETL, $CH_3NH_3PbBr_3$ as an active layer, Spiro OmeTAD as HTL, $NiO_2$ as EBL, ZnO: Al as transparent electrode/light spectrum passing layer towards DBR pairs, and DBR as a back reflector ($Si/SiO_2$).**

85.84–85.87%, and *PCE* 21.01–23.42%. However, further action for increasing the amount of active layer in the geometry of the cell results in the decline of the electrical parameters because too much high thickness causes series resistance in the cell which suppresses the electrical parameters and deteriorates the cell performing parameters. Moreover, enhancement in the thickness after the optimal value gives rise to defect state densities [40]. Thus, the obtained PV parameters are in good agreement with the lambert law.

Next, the geometry of the cell is configured with the DBR pairs which are composed of effective materials ($Si/SiO_2$) as shown in Fig 4. Four pairs of DBR bilayers are implemented to reflect the light from the bottom portion of the cell to the active region which gets absorbed in the active region and results in the enhancement of e-h pairs which helps in gaining the improved electrical parameters.

In this case, when DBR pairs are inserted, the performance parameters increase linearly because the reflected light put a positive impact on the cell and thus produces higher values. The highest photovoltaic parameters were achieved when the cell is configured with four pairs. The $V_{oc}$ climbed from 1.325–1.409V, $J_{sc}$ from 23.14–24.09 mA/cm², *FF* from 85.87–86.18% and *PCE* from 23.42–24.38% by delivering an enhancement of 0.084%, 0.95%, 0.31%, and 0.96% respectively as shown in Fig 5(A)–5(D). The recorded enhancement percentage indicates that the light is reflected in the active region, got absorbed, and gives rise to a greater number of e-h pairs which generate the enhanced PV parameters.

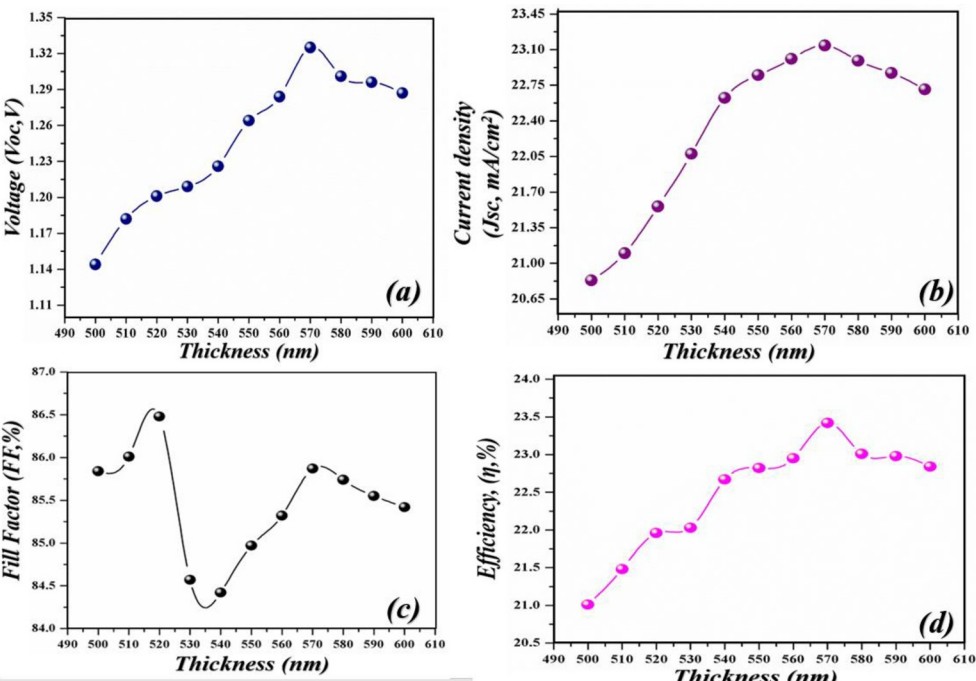

**Fig 3.** Simulated photovoltaic parameters of novel geometry (a) $V_{oc}$ as an independent function of light-harvesting layer, (b) $J_{sc}$ as an independent function of the light-harvesting layer (c) *FF* as an independent function of light-harvesting layer (d) *PCE* as an independent function of light-harvesting layer.

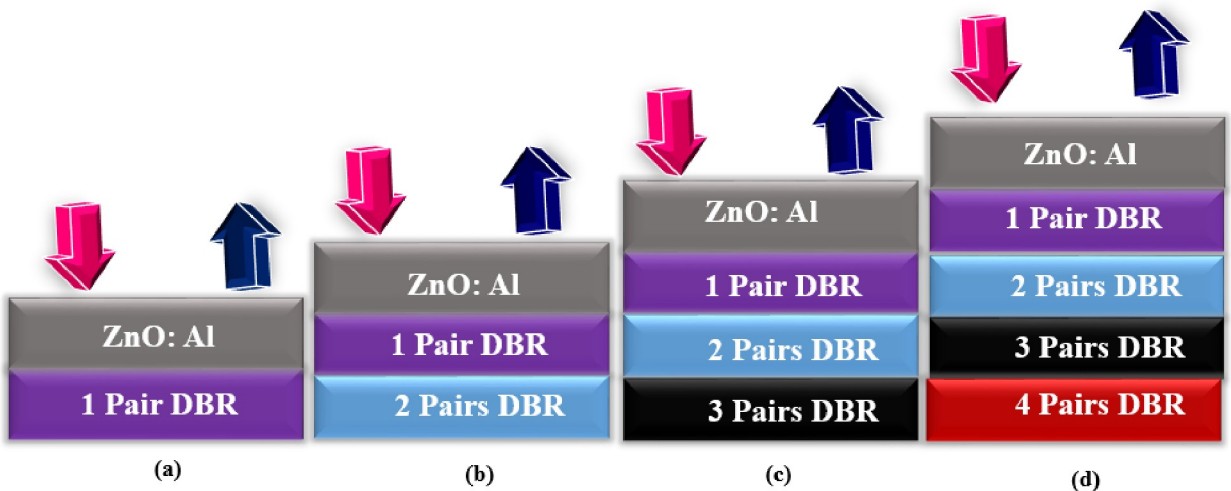

**Fig 4.** Geometry of the DBR pairs after cooperating it on the solar cell bottom area (a) single pair (b) double pair (c) three pair and (d) four pair, where each pair is composed of (Si/SiO$_2$).

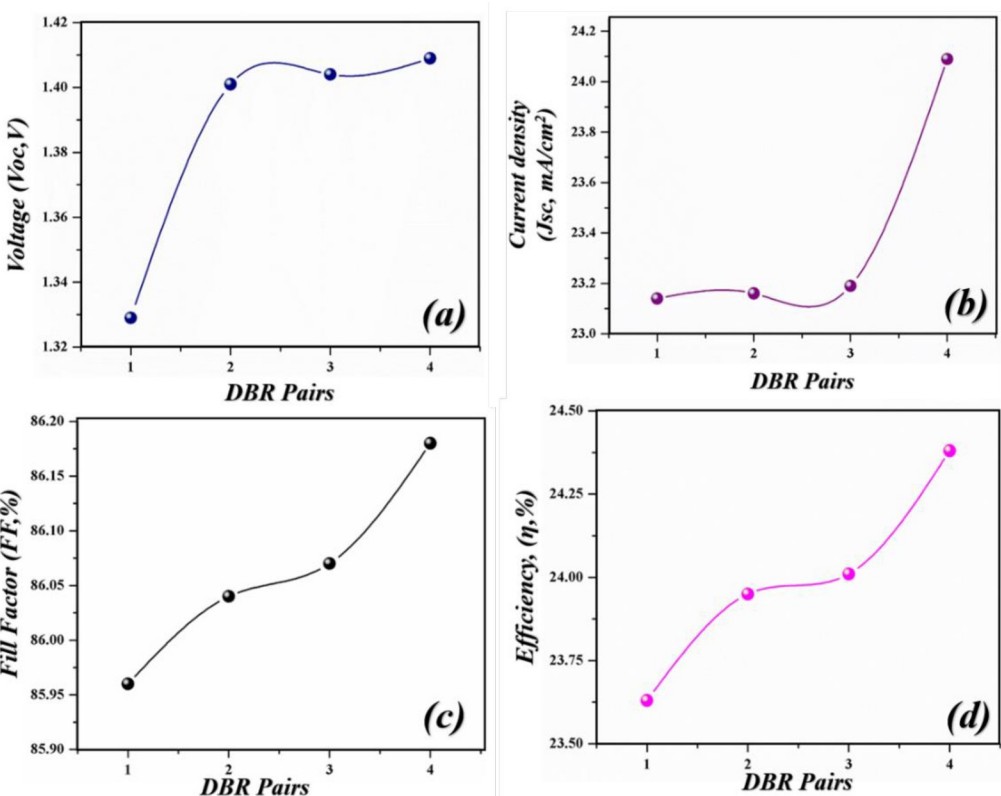

**Fig 5.** Simulated photovoltaic parameters of novel geometry with different DBR pairs (si/SnO$_2$) (a) $V_{oc}$ as an independent function of light-harvesting layer, (b) $J_{sc}$ as an independent function of a light-harvesting layer (c) *FF* as an independent function of light-harvesting layer (d) *PCE* as an independent function of light-harvesting layer.

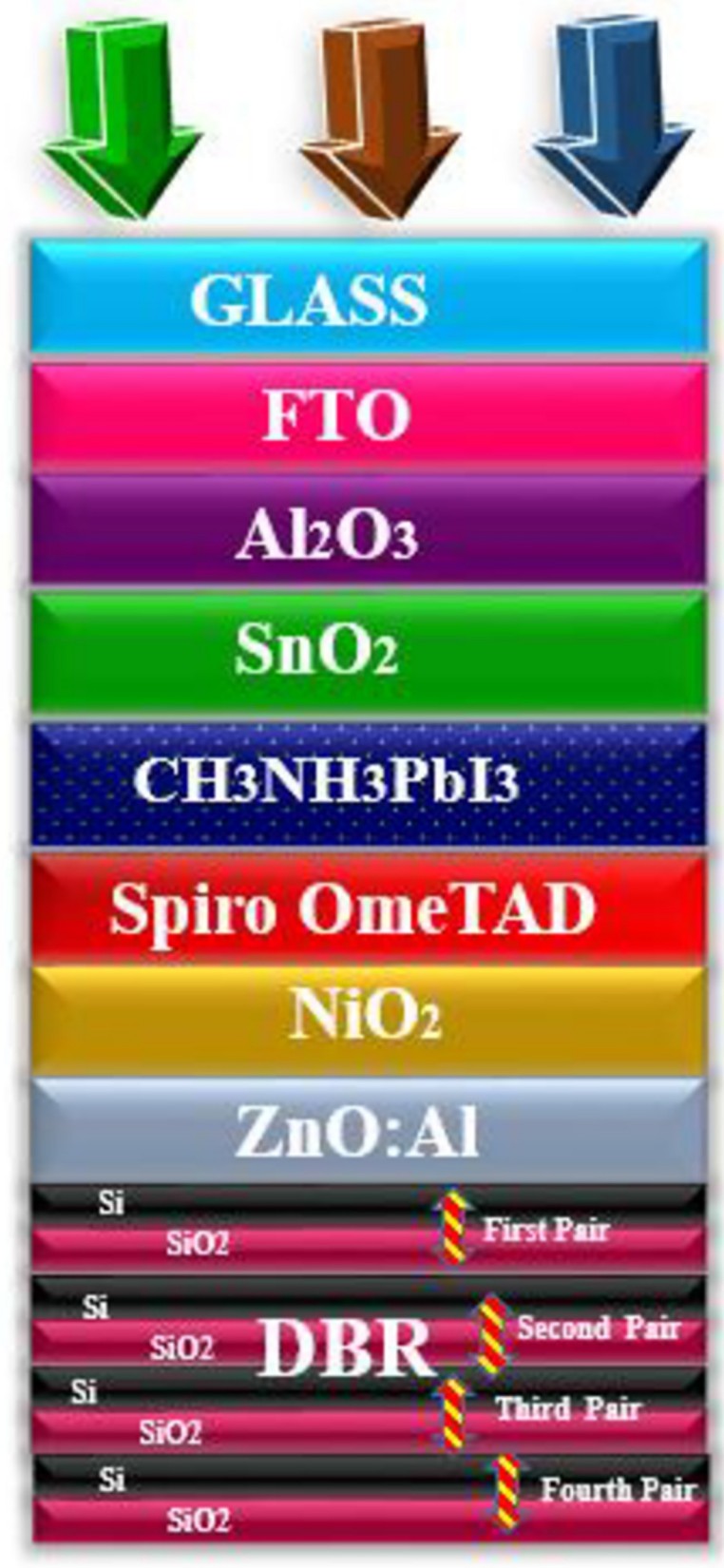

**Fig 6. Geometry of the proposed solar cell composed of different functional materials, glass as protecting layer, FTO as a top transparent electrode, Al$_2$O$_3$ as HBL, SnO$_2$ as ETL, CH$_3$NH$_3$PbI$_3$ as an active layer, Spiro OmeTAD as HTL, NiO$_2$ as EBL, ZnO: Al as transparent electrode/light spectrum passing layer towards DBR pairs, and DBR as a back reflector (Si/SiO$_2$).**

## Case 2

Next, the active material is swapped with another kind of perovskite material i.e., Methylammonium lead iodide perovskite (CH$_3$NH$_3$PbI$_3$) as displayed in Fig 6. CH$_3$NH$_3$PbI$_3$ is also a typical material that is considered while designing thin-film perovskite solar cells because it has a low and easy fabrication process [41]. Moreover, this material has also a thin-film casting ability that can be utilized in optoelectronic devices. The ability of the material to absorb a large number of photons to a good limit is because of the absorption coefficient. Thus, these advantages of the material make them an ideal candidate for thin-film technology.

In this case, the same imperative approach of modulating the active layer is considered as in the previous case. Here again, as expected the same impact of thickness modulation is perceived. The PV parameters linearly increase when the thickness of the cell is increased and after attaining the critical value of 550 nm the PV parameters started fading which suggests

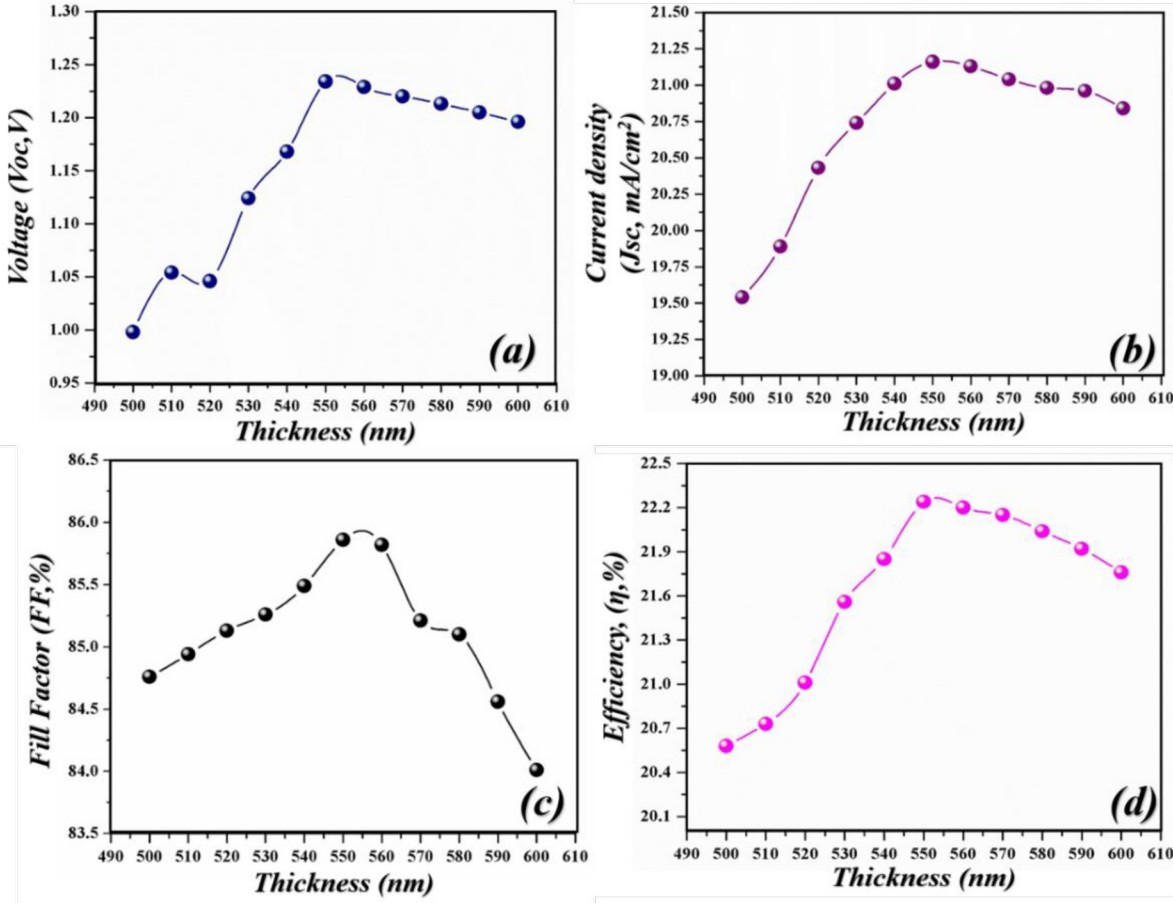

**Fig 7.** Simulated photovoltaic parameters of novel geometry (a) $V_{oc}$ as an independent function of light-harvesting layer, (b) $J_{sc}$ as an independent function of the light-harvesting layer (c) FF as an independent function of light-harvesting layer (d) PCE as an independent function of light-harvesting layer.

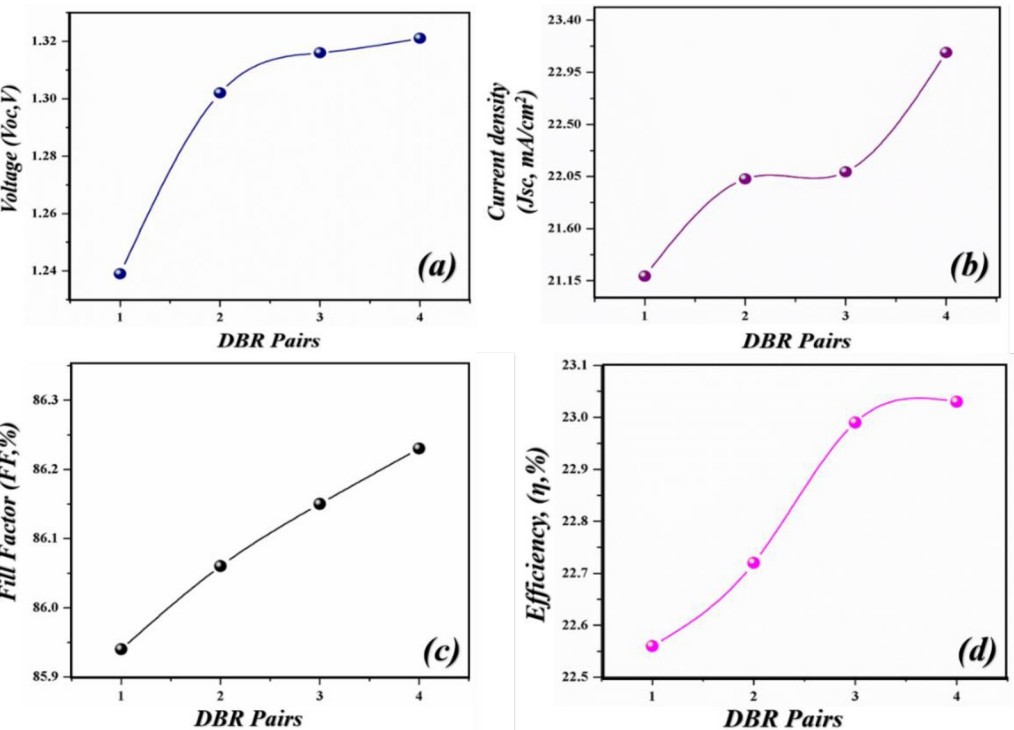

**Fig 8.** Simulated photovoltaic parameters of novel geometry with different DBR pairs (Si/SiO$_2$) (a) $V_{oc}$ as an independent function of light-harvesting layer, (b) $J_{sc}$ as an independent function of the light-harvesting layer (c) $FF$ as an independent function of light-harvesting layer (d) $PCE$ as an independent function of light-harvesting layer.

that the optimal thickness of CH$_3$NH$_3$PbI$_3$ is 550 nm in the proposed geometry. In context to the values of $V_{oc}$, $J_{sc}$, $FF$, and $PCE$ the highest attainable values are 1.23 V, 21.16 mA/cm$^2$, 85.86% and 22.24% as shown in Fig 7(A)–7(D) respectively. As discussed earlier, the fading values after the optimal thickness are obvious.

Here again, the cell is paired with DBR pairs to further improve the PV parameters. After inserting the DBR pairs, the $Voc$, $Jsc$, $FF$ and $PCE$ tends to be increased from 1.23 V, 21.16 mA/cm$^2$, 85.86%, 22.24% to 1.32 V, 23.12 mA/cm$^2$, 86.23% & 23.03% respectively as shown in Fig 8(A)–8(D). Thus, delivering an enhancement in the efficiency of 0.79%. The recorded parameters of this case are low as compared to case 1.

## Case 3

In this case, the active material is again swapped with another lead-free perovskite material as shown in Fig 9. The advantage of using lead-free based perovskite material is that it is non-toxic and environmental friendly [42]. Whereas the previous perovskite materials were toxic and non-environmental friendly because of the presence of lead. Tin-based perovskite material is also an effective material that can deliver good performance in a thin film.

The same imperative approach of thickness modulation is implemented in this geometry to observe its impact on the PV parameters. The same observation was observed in this case as well. The PV parameters improved linearly before attaining the optimal value of active thickness. At the optimal thickness, the highest efficiency of 20.67% is observed. The highest value of $V_{oc}$, $J_{sc}$, $FF$, and $PCE$ was recorded at 590 nm as shown in Fig 10(A)–10(D) respectively. As

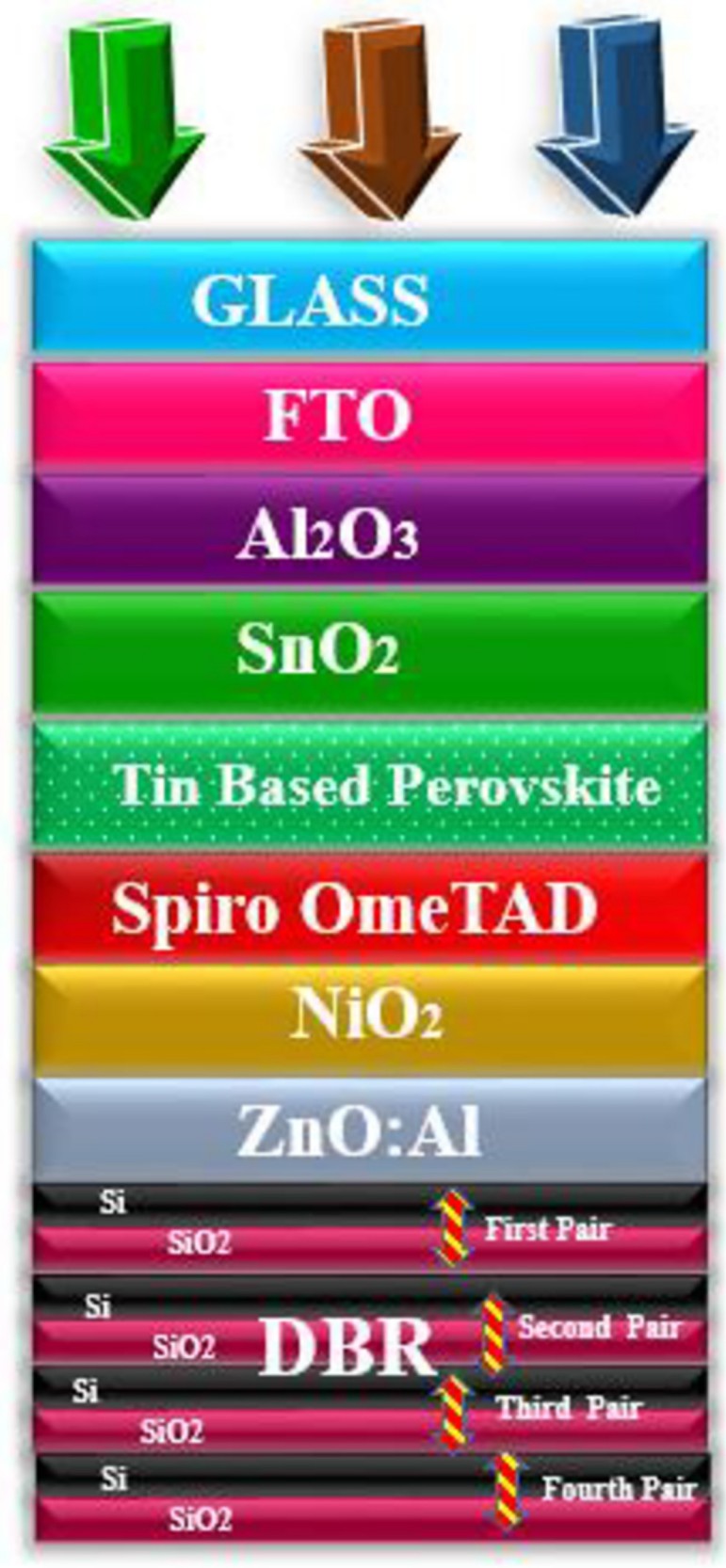

**Fig 9. Geometry of the proposed solar cell composed of different functional materials, glass as protecting layer, FTO as a top transparent electrode, Al$_2$O$_3$ as HBL, SnO$_2$ as ETL, Tin based/Lead-Free perovskite as an active layer, Spiro OmeTAD as HTL, NiO$_2$ as EBL, ZnO: Al as transparent electrode/light spectrum passing layer towards DBR pairs, and DBR as a back reflector (Si/SiO$_2$).**

discussed earlier in case 1 and case 2, the emerging decay in the values after optimal thickness is obvious.

Next, the cell is again paired with the DBR section and as observed in the previous cases, the PV parameter improved sequentially by adding the pairs of DBR. The highest efficiency of 21.28% with $V_{oc}$ = 1.168 V, $J_{sc}$ = 21.424 mA/cm$^2$ and *FF* = 86.06% were achieved with four pairs as shown in Fig 11(A)–11(D) respectively. However, the performance of this cell is low as compared to Case 1 and Case 2 but can be utilized if one has the objective to obtain energy from non-toxic and environmentally friendly materials.

Fig 12(A) demonstrates the reflectance of the structure after implementing the DBR pairs, where *N = 4* represents the number of pairs used. The absorption coverage of all the investigated structures with and without DBR is depicted in Fig 12(B), which shows that the structure with the DBR pairs sufficiently attained the high coverage of absorption as compared to the structure without DBR pairs. Furthermore and potentially the presented approach provides a new perspective towards the utilization of the DBR technique in solar cells, because the DBR pairs help in achieving the high PV performance parameters. In addition, the comparison of all the presented cases based on efficiency is summarized in Fig 12(C).

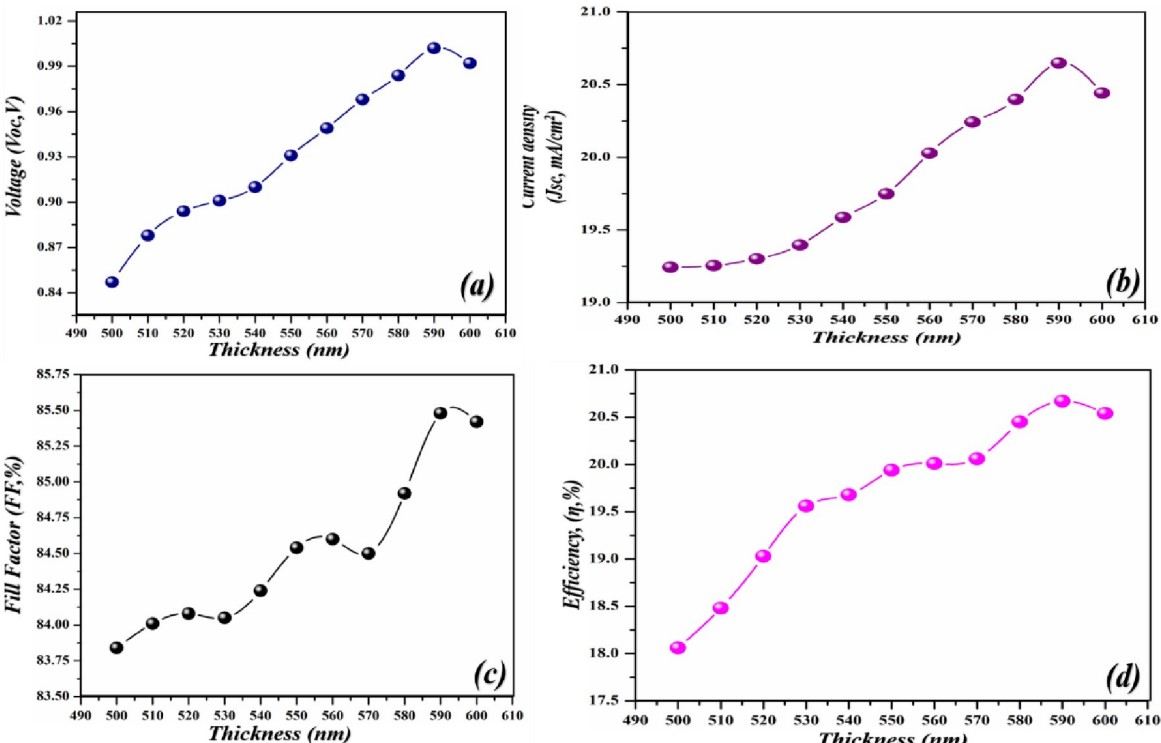

**Fig 10.** Simulated photovoltaic parameters of novel geometry (a) $V_{oc}$ as an independent function of light-harvesting layer, (b) $J_{sc}$ as an independent function of a light-harvesting layer (c) *FF* as an independent function of light-harvesting layer (d) *PCE* as an independent function of light-harvesting layer.

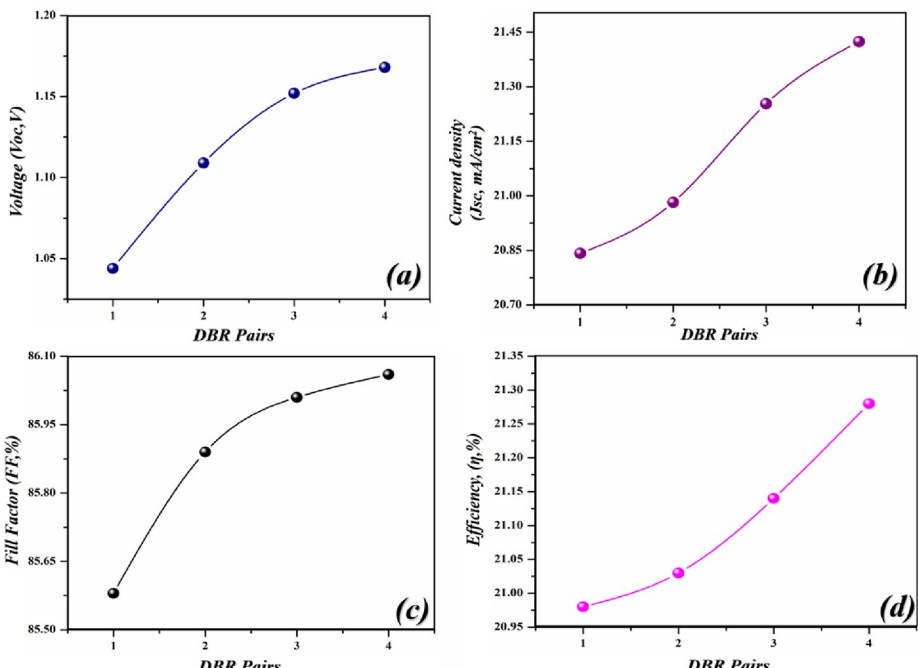

**Fig 11.** Simulated photovoltaic parameters of novel geometry with different DBR pairs ($Si/SiO_2$) (a) $V_{oc}$ as an independent function of light-harvesting layer, (b) $J_{sc}$ as an independent function of the light-harvesting layer (c) *FF* as an independent function of light-harvesting layer (d) *PCE* as an independent function of light-harvesting layer.

Eventually, a comparison is made with various kinds of other cells based on DBR and without DBR and are summarized in Table 2 & our proposed geometry holds higher conversion efficiency.

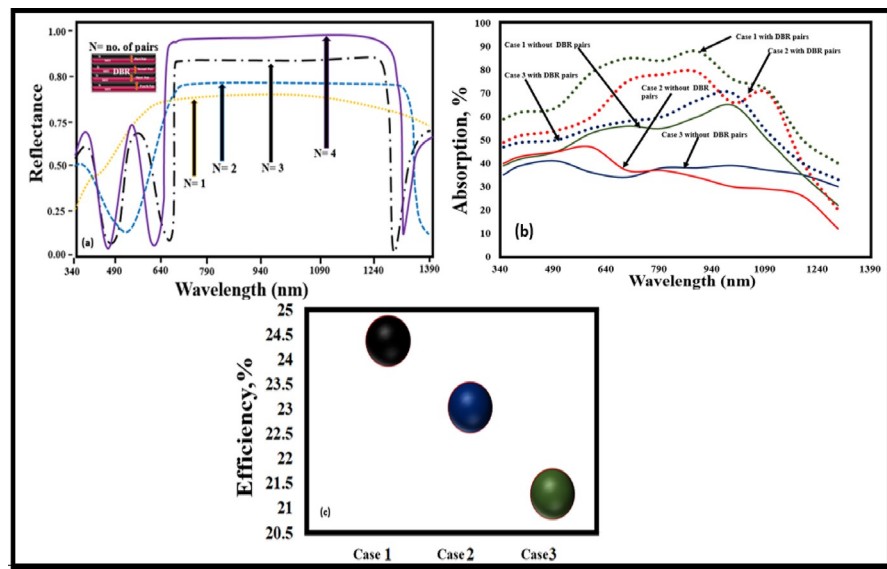

**Fig 12.** (a) Demonstration of Reflectance Vs Wavelength at optimized DBR pairs (b) Absorption coverage range of the proposed structures with and without DBR pairs (c) Optimized efficiency comparison of the proposed structures featured with DBR pairs.

**Table 2. Comparison of the current study with various kind of solar cells.**

| Reference | Structure | $\eta$ (%) |
|---|---|---|
| [17] | Glass/TCO/SnO$_2$/CdS/CdTe/Back contact | 23.01 |
| [43] | ITO/Ga-TiO2/PTB7:PC71BM/MoO3/Al | 7.72 |
| [44] | Glass/ITO/PEDOT:PSS/P3HT:PCBM/TiO$_2$/PEDOT:PSS/PTB7:PCBM/TiO$_2$/Al | 13.96 |
| [45] | SLG/Mo/CZTS/ZnS/ AZO/Ag | 3.02 |
| [46] | Glass/FTO/ZSO/Perovskite/Spiro OmeTAD/Au | 21.3 |
| [37] | FC/ARC/n-Si/p-Si/PRC/Silica nanoparticles/DBR (SiO$_2$/a:Si) | 22.54 |
| [37] | FC/ARC/n-Si/p-Si/PRC/Silica nanoparticles/ DBR (SiO$_2$/TiO$_2$) | 22.91 |
| [37] | FC/ARC/n-Si/p-Si/PRC/Silica nanoparticles/ DBR (SiO$_2$/SiNx) | 22.81 |
| [34] | Au/Spiro-MeOTAD/Perovskite/SnO$_2$/FTO/Glass/DBR (SnO$_2$/SiO$_2$) | 9.52 |
| [47] | Glass/TiO$_2$/ZnS/CIGS/ZnTe/ZnO:Al/DBR (Si/TiO$_2$) | 23.29 |
| Current Study | Glass/FTO/Al$_2$O$_3$/SnO$_2$/Perovskite/Spiro-OmeTAD/NiO$_2$/ZnO: Al/DBR(Si/SiO$_2$) | 24.38 |

## Conclusion

In summary, we numerically investigated the novel geometry of solar cells which is based on perovskite material as a photoactive layer and Si/SiO$_2$ as a back reflector in DBR pairs. The geometry of the cell-based on CH$_3$NH$_3$PbBr$_3$ with four pairs of DBR was found to be super-efficient as compared to other geometries. The proposed geometry delivers the highest efficiency of 24.38%. Moreover, the demonstrated results provide a deep insight into the geometry which can be used to capture the reflected light back into the active material more efficiently as compared to those structures which lack a back reflector.

## Supporting information

**S1 Dataset.**
(PDF)

## Author Contributions

**Conceptualization:** Waqas Farooq, Sadaqat ur Rehman.

**Data curation:** Waqas Farooq, Syed Asfandyar Ali Kazmi.

**Formal analysis:** Waqas Farooq, Syed Asfandyar Ali Kazmi, Haseeb Ahmad Khan.

**Funding acquisition:** Shanshan Tu.

**Investigation:** Waqas Farooq.

**Methodology:** Waqas Farooq, Haseeb Ahmad Khan.

**Project administration:** Sadaqat ur Rehman, Adnan Daud Khan.

**Resources:** Waqas Farooq, Syed Asfandyar Ali Kazmi.

**Software:** Waqas Farooq, Syed Asfandyar Ali Kazmi, Adnan Daud Khan.

**Supervision:** Sadaqat ur Rehman.

**Validation:** Waqas Farooq, Syed Asfandyar Ali Kazmi, Adnan Daud Khan.

**Visualization:** Waqas Farooq.

**Writing – original draft:** Waqas Farooq.

**Writing – review & editing:** Shanshan Tu, Sadaqat ur Rehman, Adnan Daud Khan, Haseeb Ahmad Khan, Muhammad Waqas, Obaid ur Rehman, Haider Ali, Muhammad Noman.

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
