## [Decision Letter · Decision Letter 0]

6 May 2021

PONE-D-21-06292

Novel Perovskite Solar Cell with Distributed Bragg Reflector

PLOS ONE

Dear Dr. Rehman,

Thank you for submitting your manuscript to PLOS ONE. After careful consideration, we feel that it has merit but does not fully meet PLOS ONE’s publication criteria as it currently stands. Therefore, we invite you to submit a revised version of the manuscript that addresses the points raised during the review process.

We look forward to receiving your revised manuscript.

Kind regards,

Jinbao Zhang

Academic Editor

PLOS ONE

Journal Requirements:

"The authors would like to thank Sarhad University of Science and Information Technology

 (SUIT), US-Pakistan Center for Advanced Studies in Energy, University of Engineering &

Technology, Peshawar, Khyber Pukhtunkhwa, Pakistan, University of Engineering and

Technology, Mardan, Khyber Pukhtunkhwa, Pakistan,and University of Technology,

Nowshera, Pakistan for the support provided for this research."

"This work was supported by The China National Key R&D Program (No. 2018YFB0803600), Natural Science Foundation of China(61801008), Beijing Natural Science Foundation (L172049), Scientific Research Common Program of Beijing Municipal Commission of Education (No.KM201910005025) and Chinese Postdoctoral Science Foundation (No. 2020M670074)."

5. Please ensure that you refer to Figure 9 and Figure 11 in your text as, if accepted, production will need this reference to link the reader to the figure.

Reviewers' comments:

Reviewer's Responses to Questions

**Comments to the Author**

1. Is the manuscript technically sound, and do the data support the conclusions?

Reviewer #1: Partly

Reviewer #2: Partly

2. Has the statistical analysis been performed appropriately and rigorously? 

Reviewer #1: No

Reviewer #2: Yes

3. Have the authors made all data underlying the findings in their manuscript fully available?

Reviewer #1: Yes

Reviewer #2: Yes

4. Is the manuscript presented in an intelligible fashion and written in standard English?

Reviewer #1: No

Reviewer #2: No

5. Review Comments to the Author

Reviewer #1: The manuscript is trying to report the numerically investigated perovskite based solar cells with DBR structures, which is an interesting area. but the logical structure, the data and languages of the manuscript need to be improved before consideration for publication. the comments are listed below:

1. The three perovskite materials structures should be compared side by side to draw the conclusion.

2. The performance (e.g. reflectance) of DBRs should be provided. Also, the authors should be clarify the DBR materials, Si/SiO2 or Si/SnO2?

3. The properties of three perovskite materials should be provided, including structures, optical properties, etc.

4. there are some mislabels and lots of highlights, e.g. si/siO2 should be Si/SiO2? please check the whole manuscript carefully to avoid them.

Reviewer #2: This work modelled the performance of CH3NH3PbI3, CH3NH3PbBr3, and CH3NH3SnI3 perovskite cells with DBR as the rear reflector. It has certain values, but substantial improvements are required before publication. Specific comments are:

1. The language needs to be improved.

2. Line 60-74, this paragraph needs a bit more logic. The authors are just listing various works, but these works are not relevant to the topic of your work. I would suggest either discuss the improvement of perovskite efficiencies over the years, or survey the use of various rear reflectors.

3. Check equation 7

4. Please double check you DBR materials, Si/SiO2 or Si/SnO2? Si/SnO2 in used in all figure captions.

5. Line 134, “The cell performance surprisingly increases with increasing the height of the active layer.” Why suprisingly? Normally we use “thickness” instead of “height” to describe layer thickness.

6. Line 141, “supplementary Fig 3(a), (b), (c), and (d) respectively” there is no supplementary information, delete supplementary.

7. Line 155 “Four pairs of DBR”, DBR refers to the whole alternating layer stack, a more proper way should be four pairs of DBR bilayers.

8. The comparison in Table 2 does not provide any useful information. The authors are comparing totally irrelevant designs. The audience could nor draw any useful information from the comparison between different structures, especially experimental work vs. modelling results on various solar cell. A more appropriate comparison table could be comparing the same perovskite structure with various rear reflectors designs, so the authors could conclude whether a DBR is superior than other reflectors.

6. PLOS authors have the option to publish the peer review history of their article (what does this mean?). If published, this will include your full peer review and any attached files.

Reviewer #1: No

Reviewer #2: No

---

## [Author Response · Author response to Decision Letter 0]

14 Jun 2021

Response to Reviewers’ Comments 

(PONE-D-21-06292)

Dear Editor and Reviewers: 

I, as a corresponding author on behalf of all coauthors, would like to thank you for the careful review and constructive comments regarding the initial version of our manuscript, which was submitted to PlosOne Journal entitled “Novel Perovskite Solar Cell with Distributed Bragg Reflector” (Manuscript ID: PONE-D-21-06292). We would also like to express our gratitude for providing us with constructive comments that turned out to improve the quality of the revised version of the paper. We carefully reviewed our paper considering reviewers’ comments and revised the paper accordingly. Moreover, proofreading has been carried out thoroughly and extensively by removing all the presentation errors and ambiguities in order to improve the readability of the paper. We hope that the reviewers find the changes satisfactory and the revised manuscript successfully addresses the questions and comments of the reviewers. We attach this authors’ reply letter that discusses our changes made with respect to each of the reviewers’ comments. The detailed response to each comment and our corresponding revisions are outlined on the next page (reviewers’ comments in black, our replies in blue). For convenience, new additions and major changes are highlighted in “YELLOW” color in the revised manuscript. Again, the reviews were very constructive, and the comments have been very helpful in terms of improving our work. We take this opportunity to express our appreciation for your expertise and invaluable assistance in reviewing our draft. We hope that the revised draft provides a better presentation of our work. Should you have any questions, please do not hesitate to contact us. 

Respectfully yours, 

Shanshan Tu, 

Ph.D. 

E-mail: sstu@bjut.edu.cn

Response to Reviewer Comments

(Reviewer 1)

Reviewer 1: The three perovskite materials structures should be compared side by side to draw the conclusion.

Author Reply: We thank the respected Reviewer for this comment as side-by-side comparison is always considered best for better understanding of the structure performance. The point is valid, and comparison graph has been added in the revised version of the manuscript such as side by side comparison of the investigated on the basis of Absorption and efficiency. Please visit Figure 12 (b) and (c). 

Reviewer 1: The performance (e.g. reflectance) of DBRs should be provided. Also, the authors should be clarify the DBR materials, Si/SiO2 or Si/SnO2?

Author Reply: We thank the respected Reviewer for this comment. The reflectance graph is important for understanding the DBR performance. The reflectance graph has been added in the revised version of the manuscript. Please visit Figure 12 (a). For the DBR material correction, we are once again thanking to the Reviewer for observing typo error. The utilized material for DBR is Si/SiO2.

Reviewer 1: The properties of three perovskite materials should be provided, including structures, optical properties, etc

Author Reply: We thank the respected Reviewer for this comment. The properties of the materials are available in Table 1. And for the optical properties, we have added the absorption (%) of all the structures with and without DBR pairs. The included graph helps in better understanding of the DBR pairs and its impact on the investigated structures. Please visit Figure 12 (b). 

Reviewer 1: There are some mislabels and lots of highlights, e.g. si/siO2 should be Si/SiO2? Please check the whole manuscript carefully to avoid them.

Author Reply: We really want to thank the respected Reviewer for highlighting the mislabels. The correction has been performed in the revised version of the manuscript. 

Response to Reviewer Comments

(Reviewer 2)

Reviewer 2: The language needs to be improved.

Author Reply: We thank the respected Reviewer for this comment. Language is an important tool for understanding the manuscript. The point is valid, and we have improved the language of the manuscript. 

Reviewer 2: Line 60-74, this paragraph needs a bit more logic. The authors are just listing various works, but these works are not relevant to the topic of your work. I would suggest either discuss the improvement of perovskite efficiencies over the years, or survey the use of various rear reflectors.

Author Reply: We thank the respected Reviewer for pointing out the weaker area of the manuscript. The point is valid as well. The literature regarding the rear reflectors has been added in the revised version of the manuscript. Please visit line 74-82. 

Reviewer 2: Check equation 7

Author Reply: Respected Reviewer, we have checked Equation 7, it is used for calculating the power conversion efficiency (PCE), ղ of the cell and is correct. 

Reviewer 2: Please double check you DBR materials, Si/SiO2 or Si/SnO2? Si/SnO2 in used in all figure captions.

Author Reply: We really want to thank respected Reviewer of pointing out this error. The point is valid, and we have checked the DBR material. It was typing error and has been corrected in the revised version of the manuscript. The utilized material for DBR is Si/SiO2.

Reviewer 2: Line 134, “The cell performance surprisingly increases with increasing the height of the active layer.” Why suprisingly? Normally we use “thickness” instead of “height” to describe layer thickness

Author Reply: We thank the respected Reviewer for correcting us. The concern is valid, and the word “height” is replaced with the more appropriate word “thickness”. 

Reviewer 2: Line 141, “supplementary Fig 3(a), (b), (c), and (d) respectively” there is no supplementary information, delete supplementary.

Author Reply: We thank the respected Reviewer for this comment. The word supplementary has been removed as per respected suggestion. 

Reviewer 2: Line 155 “Four pairs of DBR”, DBR refers to the whole alternating layer stack, a more proper way should be four pairs of DBR bilayers.

Author Reply: We really want to thank the respected Reviewer for this comment. As per respected suggestion the “Four pairs of DBR” is replaced with the more appropriate word “ Four pairs of DBR bilayers”. 

Reviewer 2: The comparison in Table 2 does not provide any useful information. The authors are comparing totally irrelevant designs. The audience could nor draw any useful information from the comparison between different structures, especially experimental work vs. modelling results on various solar cell. A more appropriate comparison table could be comparing the same perovskite structure with various rear reflectors designs, so the authors could conclude whether a DBR is superior than other reflectors.

Author Reply: We thank the respected Reviewer for this comment. The comparison table was made to understand the performance of different kind of solar cells with the investigated solar cell material as there is no such material available in the literature for perovskite incorporated with DBR and here our novelty stands. 

However, as per respected suggestion by the Reviewer, we have updated the comparison table 2 by adding more appropriate structures based on DBR.

---

## [Decision Letter · Decision Letter 1]

21 Jul 2021

PONE-D-21-06292R1

Novel Perovskite Solar Cell with Distributed Bragg Reflector

PLOS ONE

Dear Dr. Rehman,

Thank you for submitting your manuscript to PLOS ONE. After careful consideration, we feel that it has merit but does not fully meet PLOS ONE’s publication criteria as it currently stands. Therefore, we invite you to submit a revised version of the manuscript that addresses the points raised during the review process.

We look forward to receiving your revised manuscript.

Kind regards,

Jinbao Zhang

Academic Editor

PLOS ONE

Journal Requirements:

Reviewers' comments:

Reviewer's Responses to Questions

**Comments to the Author**

1. If the authors have adequately addressed your comments raised in a previous round of review and you feel that this manuscript is now acceptable for publication, you may indicate that here to bypass the “Comments to the Author” section, enter your conflict of interest statement in the “Confidential to Editor” section, and submit your "Accept" recommendation.

Reviewer #1: All comments have been addressed

Reviewer #2: All comments have been addressed

2. Is the manuscript technically sound, and do the data support the conclusions?

Reviewer #1: Yes

Reviewer #2: Yes

3. Has the statistical analysis been performed appropriately and rigorously? 

Reviewer #1: Yes

Reviewer #2: Yes

4. Have the authors made all data underlying the findings in their manuscript fully available?

Reviewer #1: Yes

Reviewer #2: Yes

5. Is the manuscript presented in an intelligible fashion and written in standard English?

Reviewer #1: Yes

Reviewer #2: Yes

6. Review Comments to the Author

Reviewer #1: The authors have been addressed my questions, the manuscript could be considered to publish after revising the following points:

1. In Figure 4, DBR1,DBR2,DBR3, DBR4 should be replaced with 1 pair DBR, 2 pairs DBR, 3 pairs DBR,4 pairs DBR.

2. In Figrue 12a and b, there are no labels for the lines. The different color lines should be labelled clearly.

Reviewer #2: The manuscript has been improved, but there are same parts need further clarification.

1. Equation 7, the symbol for efficiency is not shown properly. I am seeing a question mark inside a box.

2. Figure 5, the lines connecting the dots should not be smooth lines. Fig5 (b), the smooth line indicates 2.5 DBR pairs is worse than 2 pairs.

3. Figure 12, legend is need for plots (a) and (b). what does each coloured lines mean?

7. PLOS authors have the option to publish the peer review history of their article (what does this mean?). If published, this will include your full peer review and any attached files.

Reviewer #1: No

Reviewer #2: No

---

## [Author Response · Author response to Decision Letter 1]

27 Jul 2021

A step by step response to reviewers is attached.

---

## [Editor Report · Decision Letter 2]

23 Aug 2021

PONE-D-21-06292R2

Novel Perovskite Solar Cell with Distributed Bragg Reflector

PLOS ONE

Dear Dr. Rehman,

Thank you for submitting your manuscript to PLOS ONE. After careful consideration, we feel that it has merit but does not fully meet PLOS ONE’s publication criteria as it currently stands. Therefore, we invite you to submit a revised version of the manuscript that addresses the points raised during the review process.

We look forward to receiving your revised manuscript.

Kind regards,

Jinbao Zhang

Academic Editor

PLOS ONE

Journal Requirements:

Additional Editor Comments (if provided):

Please response the comments below from reviewers.

1.The authors have been addressed my questions, the manuscript could be considered to publish after revising the following points:

(1). In Figure 4, DBR1,DBR2,DBR3, DBR4 should be replaced with 1 pair DBR, 2 pairs DBR, 3 pairs DBR,4 pairs DBR.

(2). In Figrue 12a and b, there are no labels for the lines. The different color lines should be labelled clearly.

2. The manuscript has been improved, but there are same parts need further clarification.

(1). Equation 7, the symbol for efficiency is not shown properly. I am seeing a question mark inside a box.

(2). Figure 5, the lines connecting the dots should not be smooth lines. Fig5 (b), the smooth line indicates 2.5 DBR pairs is worse than 2 pairs.

(3). Figure 12, legend is need for plots (a) and (b). what does each coloured lines mean?
---

## [Author Response · Author response to Decision Letter 2]

6 Oct 2021

Response to Reviewer Comments

(Reviewer 1)

Reviewer 1: The authors have been addressed my questions, the manuscript could be considered to publish after revising the following points:

Author Reply: We thank the respected Reviewer for accepting our efforts and suggesting acceptance after minor revisions. 

Reviewer 1: 1. In Figure 4, DBR1,DBR2,DBR3, DBR4 should be replaced with 1 pair DBR, 2 pairs DBR, 3 pairs DBR,4 pairs DBR.

Author Reply: We thank the respected Reviewer for this comment, as these kinds of comments always make the manuscript more worthy and easy for the readers to understand the study more precisely and accurately. As per respected suggestion, Figure 4 has been updated in the revised version of the manuscript. 

Reviewer 1: 2. In Figure 12a and b, there are no labels for the lines. The different color lines should be labelled clearly.

Author Reply: We thank the respected Reviewer for comment. The point is valid, and proper labels for the colored lines have been updated in the revised version of the manuscript. 

Response to Reviewer Comments

(Reviewer 2)

Reviewer 2: The manuscript has been improved, but there are same parts need further clarification.

Author Reply: We thank the respected Reviewer for observing improvement in the manuscript. 

Reviewer 2: Equation 7, the symbol for efficiency is not shown properly. I am seeing a question mark inside a box

Author Reply: Respected Reviewer, the symbol of efficiency (ղ) in Equation 7 is clearly visible in our revised version. There might be some internal error, which is not showing the correct symbol to the Reviewers draft. Once again, we have edited Equation 7, and hope so, it will be visible clearly in the revised version of the manuscript. 

Reviewer 2: Figure 5, the lines connecting the dots should not be smooth lines. Fig5 (b), the smooth line indicates 2.5 DBR pairs is worse than 2 pairs.

Author Reply: We respect the Reviewer's point of view regarding Figure 5 (b). The plot is made in the latest version of the Origin PRO, and the B-Spine option is used for the presented Data. There are no 2.5 pairs used. The number of DBR is as follows 1, 2, 3, and 4. The slight decay between 2 and 3 pairs is the slope that software has produced to give a smooth graphical line. 

Reviewer 2: Figure 12, legend is need for plots (a) and (b). what does each coloured lines mean?

Author Reply: We thank the respected Reviewer for this comment. The point is valid and Figures 12 (a) and (b) have been updated in the revised version of the manuscript. Colored lines represent different cases of the study. 

---

## [Editor Report · Decision Letter 3]

27 Oct 2021

Novel Perovskite Solar Cell with Distributed Bragg Reflector

PONE-D-21-06292R3

Dear Dr. Rehman,

We’re pleased to inform you that your manuscript has been judged scientifically suitable for publication and will be formally accepted for publication once it meets all outstanding technical requirements.

Kind regards,

Jinbao Zhang

Academic Editor

PLOS ONE
---

## [Editor Report · Acceptance letter]

17 Nov 2021

PONE-D-21-06292R3 

Novel Perovskite Solar Cell with Distributed Bragg Reflector 

Dear Dr. Rehman:

I'm pleased to inform you that your manuscript has been deemed suitable for publication in PLOS ONE. Congratulations! Your manuscript is now with our production department. 

Kind regards, 

on behalf of

Dr. Jinbao Zhang 

Academic Editor

PLOS ONE